# Prediction of the Comprehensive Error Field in the Machining Space of the Five-Axis Machine Tool Based on the "S"-Shaped Specimen Family

**Shi Wu \*** , **Zeyu Dong** , **Fei Qi and Zhendong Fan**

Key Laboratory of Advanced Manufacturing and Intelligent Technology, Ministry of Education, Harbin University of Science and Technology, Harbin 150080, China; 18845041407@163.com (Z.D.); qifei368@163.com (F.Q.); fzd1778417460@163.com (Z.F.)
\* Correspondence: swu@hrbust.edu.cn; Tel.: +86-187-4568-7640

**Abstract:** In order to quickly and accurately predict the spatial geometric error field of the five-axis machine tool processing, a method for predicting the comprehensive error field of the five-axis machine tool processing space based on the "S"-shaped specimen family is studied. Firstly, for the five-axis CNC machine tool in the form of A-C dual turntable, the geometric error model of the rotating axis is established based on the multi-body dynamics theory; the error mapping relationship between the processing technology system and the workpiece is analyzed based on the "S"-shaped specimen family, and the identification of 12 geometric errors of the two rotating shafts. Then, the error value of the sampling point is measured based on the "S"-shaped test piece in machine contact, and the double-circle center coordinate value is determined according to the curvature of the measured wire of the test piece, in order to identify the geometric errors of the two rotation axes of the five-axis machine tool. Finally, based on the prediction method, the comprehensive error field of the five-axis CNC machine tool processing space is analyzed. Compared with other geometric error identification methods, the measurement accuracy of this method meets the processing requirements and can further evaluate the comprehensive performance of the machine tool.

**Keywords:** five-axis CNC machine tool; "S"-shaped sample; machining error; spatial error field

## 1. Introduction

The continuous development of processing technology has contributed toward the widespread use of five-axis CNC machine tools in the processing of complex parts. Five-axis CNC machine tools involve many errors in the processing of such parts. These errors can be categorised into thermal errors, geometric errors, dynamic cutting force errors, and servo control errors.

Geometric errors account for a large proportion of these errors. They are mainly caused by issues in the process of machine tool part manufacturing and assembly. There are many geometric errors in machine tools, and research on such errors mainly focuses on their modelling, measurement, and identification.

Various methods are adopted for modelling geometric errors in machine tools. Currently, geometric error modelling is mainly based on multi-body system theory, which has the characteristics of high versatility and accuracy. Qiao et al. proposed a new calibration model based on the exponential product rotation theory of five-axis machine position-independent geometric errors (PIGEs), which requires only four independent parameters of the rotating axis and two independent parameters of the moving axis; thus, dimensional reduction of the recognition coefficient matrix can reduce the amount of calculation to a certain extent [1]. Zhong et al. regarded a multi-body system. They established a model of the geometric error related to the position of the axis of rotation and proposed an improved virtual rigid body recognition method to calculate the positioning error of the moving

axis [2]. Fan et al. established a five-axis machine tool by means of the alignment coordinate transformation matrix and proposed a method for predicting and recognising the geometric error of the five-axis machine tool rotation axis, which uses the geometric profile error to link the geometric error with the tolerance. This approach can predict the geometric error of the rotating axis more accurately [3]. Fu et al. employed the product of exponentials (POE) theory to obtain the error vector component of the position-independent error, established the error contribution value of each axis according to the differential change between the coordinate systems, derived the error sensitivity matrix of each axis, and finally modelled the geometric error contribution of each axis of the five-axis machine tool [4]. Wu et al. proposed a comprehensive prediction and compensation method for the geometric error caused by the flat shaft positioning error of a multi-axis machine. In this method, the alignment transformation matrix (HTM) and multi-body system (MBS) theory are used to establish a positioning error model that only considers the moving axis [5].

Methods for measuring machine geometry errors are generally categorised into direct and indirect measurement methods. Direct measurement methods are generally used in measuring equipment such as laser interferometers. Indirect measurement methods can be used for indirect geometric error identification with test pieces. The five-axis CNC machine rotary axis can be measured using a club instrument. First, they used the club instrument to measure the geometric error of the machine via three different measurement methods. Then, they established the decoupling model of the geometric error parameters by means of the alignment coordinate transformation matrix. Finally, they identified the geometric error parameters of the rotating axis of the machine tool [6]. This tool uses lasers to measure the geometric error of the three straight axes of a three-axis machine tool or five-axis machine, which is more efficient than conventional laser interferometer measurement owing to the ease of installation of the measuring system [7]. Guo et al. proposed a continuous measurement calibration method for measuring the five-axis machine rotary shaft position-related geometric error (PDGE), which reduces the number of installation adjustments of the club meter (DBB) in each measurement, thereby minimising the installation error of the measuring equipment. In this approach, the PDGEs are based on the adaptive least absolute shrink selection operator (LASSO). Identification methods have also been proposed to improve the recognition accuracy [8]. From the viewpoint of machining tests, Huang et al. studied a method for identifying the position-related geometric error and position installation error of the five-axis machine tool. They considered the effect of the machine's geometric error on the workpiece clamp and determined the geometric error (PIGEs) independent of the position of the rotary shaft (PIGEs) using impeller machining parts and the three-coordinate measuring machine (CMM) [9]. Using the machine measurement to identify the geometric error of the rotating axis and the error caused by the tool setting, which can estimate the geometric error from the rotating shaft to the tool holder and improve the precision of the processing by compensating the error component [10].

The "S"-shape for the five-axis CNC machine tool has officially become an international standard. "S"-shaped test pieces are used to comprehensively evaluate the machining performance of five-axis machine tools. Many researchers have conducted numerous studies in this regard [11]. Jiang et al. analysed the "S"-shaped specimen based on the S-shaped sample and determined the reason for the abnormal surface processing pattern of the "S"-shaped specimen by establishing the simulation platform of the servo system. They concluded that the "S"-shaped specimen can better demonstrate the dynamic performance of the five-axis machine tool servo system [12]. Wang et al. proposed an analysis method for local and global errors based on the S-shaped test piece; decoupled the key geometric error parameters that cause the contour error of the test piece; linked the geometric characteristics of the test piece, the geometric error of the machine tool, and the defects in the specimen processing; and mastered the degree and law of the influence of the geometric error on the space error of the test piece [13]. In addition, an optimised numerical method for straight-line side milling was proposed, which considerably reduces the calculation time required for tool positioning [14]. Another study considered the "S"-shaped test piece, namely, the

optimised single-point offset (OSPO) method, so as to reduce the error effect when the processing accuracy of multi-axis machine tools is comprehensively detected. Compared to the traditional single-point offset (SPO) method, this method reduces the theoretical errors more effectively [15]. Some researchers used the standard sphere measurement method to obtain 10 average positional errors of the two rotating axes by inputting them into the coordinate system [16–18]. Compared with the traditional measurement method, the accuracy of this method is about 91.8%, but the cost is lower, and the measurement time is shorter [19].

In this paper, a five-axis CNC machine tool based on the "S"-shaped specimen family is proposed to identify the geometric error of the rotating shaft of the machine tool. First, the geometric error models in the ideal and actual states of the rotating axis of the machine are established by means of the alignment coordinate transformation matrix. Then, contact measurement is performed using the "S"-shaped test family at different rotation angles, and the A-axis and C-axis are analysed to identify the values of the 12 geometric error elements of the two rotating axes. Thus, the spatial comprehensive error field of the machine processing space is obtained.

## 2. Rotating Axis Geometry Error Model

The structure of the five-axis CNC machine, which is based on the A-C dual turntable, is shown in Figure 1. With the bed as the demarcation, the A and C axes are on the workpiece side and the *X*, *Y*, and *Z* axes are on the tool side. To facilitate the description of the movement of the machine, we establish the coordinate system as shown in Figure 2.

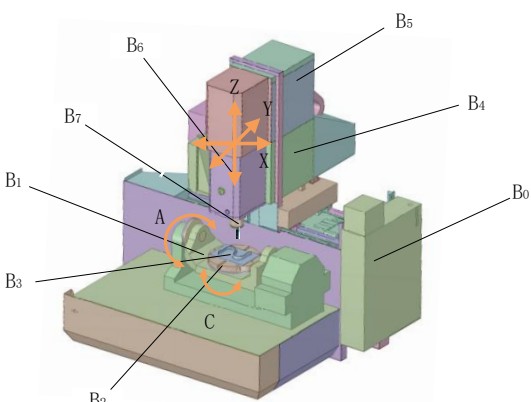

**Figure 1.** A–C dual turntable five-axis CNC machine structure. $B_0$-Bed body, $B_1$-A axis, $B_2$-C axis, $B_3$-S-shaped specimen, $B_4$-X axis, $B_5$-Y axis, $B_6$-Z axis, and $B_7$-Tool.

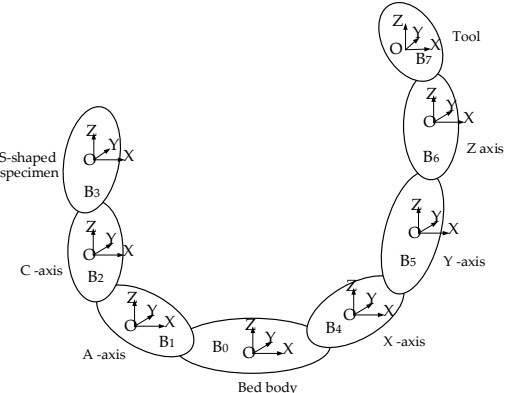

**Figure 2.** Machine topology and coordinate diagram. $B_0$-Bed body, $B_1$-A axis, $B_2$-C axis, $B_3$-S-shaped specimen, $B_4$-X axis, $B_5$-Y axis, $B_6$-Z axis, and $B_7$-Tool.

### 2.1. Rotating Axis Geometric Error Element

The geometric error elements of the five-axis CNC machine rotation shaft include three movement errors and three corner errors. Considering the A-C dual-turning five-axis CNC machine as an example, as shown in Table 1, there are 12 error elements of the two rotating axes A and C. The position definitions of the various geometric error elements of the A-axis are described in Figure 3, and the C-axis is not repeated.

**Table 1.** Various geometric error elements of A and C axes.

| Shaft | Corner Error | | | Movement Error | | |
|---|---|---|---|---|---|---|
| A-axis | $\varepsilon_{\alpha A}$ | $\varepsilon_{\beta A}$ | $\varepsilon_{\gamma A}$ | $\delta_{xA}$ | $\delta_{yA}$ | $\delta_{zA}$ |
| C-axis | $\varepsilon_{\alpha C}$ | $\varepsilon_{\beta C}$ | $\varepsilon_{\gamma C}$ | $\delta_{xC}$ | $\delta_{yC}$ | $\delta_{zC}$ |

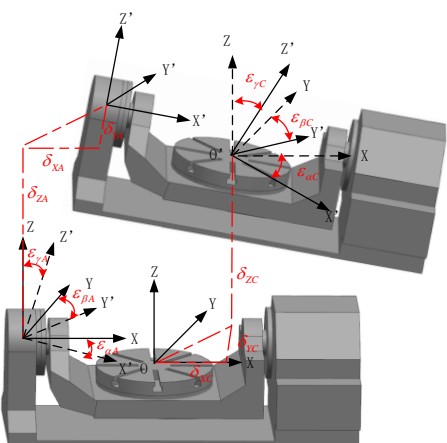

**Figure 3.** A and C axes' geometric error elements.

### 2.2. Geometric Model in The Ideal State of the Axis of Rotation

According to the coordinate conversion relationship between the motion axes of the machine, the position coordinates of the tip points in the three directions of the set workpiece coordinate system (WCS) are represented by *X*, *Y*, and *Z*, while *I*, *J*, and *K* represent the position vector of the tool in the WCS, and the position expression of the tip points in the ideal state is established. The tool attitude $p = (X, Y, Z, 1)^T$ expression: $s = (I, J, K, 0)^T$

$$p_1 = [T(x) \cdot T(y) \cdot T(z)]^{-1} \cdot R(a) \cdot R(c) \cdot p_2 \qquad (1)$$

$$v_1 = R(a) \cdot R(c) \cdot v_2 \qquad (2)$$

where: $T(x) \cdot T(y) \cdot T(z) = \begin{bmatrix} 1 & 0 & 0 & x \\ 0 & 1 & 0 & y \\ 0 & 0 & 1 & z \\ 0 & 0 & 0 & 1 \end{bmatrix}$, $R(a) = \begin{bmatrix} 1 & 0 & 0 & 0 \\ 0 & cos(a) & -sin(a) & 0 \\ 0 & sin(a) & cos(a) & 0 \\ 0 & 0 & 0 & 1 \end{bmatrix}$,

$R(c) = \begin{bmatrix} cos(c) & -sin(c) & 0 & 0 \\ sin(c) & cos(c) & 0 & 0 \\ 0 & 0 & 1 & 0 \\ 0 & 0 & 0 & 1 \end{bmatrix}$.

When the position $p_1 = (0, 0, 0, 1)^T$ of the starting tip point, the attitude of the starting tool, $v_1 = (0, 0, 1, 0)^T$ after the coordinate transformation to get the ideal position and attitude of the tool:

$$p = R(-c) \cdot R(-a) \cdot T(x) \cdot T(y) \cdot T(z) \cdot p_1 \qquad (3)$$

$$v = R(-c) \cdot R(-a) \cdot v_1 \qquad (4)$$

Combining (3) and (4), we get

$$
\begin{cases}
X = x \cdot \cos c + y \cdot \sin c \cdot \cos a + z \cdot \sin c \cdot \sin a \\
Y = -x \cdot \sin c + y \cdot \cos c \cdot \cos a + z \cdot \cos c \cdot \sin a \\
Z = -y \cdot \sin a + z \cdot \cos a \\
I = \sin c \cdot \sin a \\
J = \cos c \cdot \sin a \\
K = \cos a
\end{cases}
\tag{5}
$$

Equation (5) reflects the position and attitude of the tool in the ideal (error-free) state in the established WCS, as can be seen from Equation (5), the position of the tip point $X$, $Y$, $Z$, and the attitude of the tool $I$, $J$, $K$ is about the straight axis translation $x$, Functions of $y$, $z$, and rotation axis rotations $a$, $c$.

### 2.3. Geometric Error Model in The Actual State of The Axis of Rotation

In the actual state, $X$, $Y$, and $Z$ denote the actual position coordinates of the tool tip point in the three directions in the WCS, while $I'$, $J'$, and $K'$ denote the actual posture vector of the tool in the WCS. Owing to the existence of the geometric error of the rotation axis, the transformation matrix of the actual position and posture of the tool in the WCS is obtained by multiplying the error transformation matrix and the two rotation axes on an error-free basis, as shown in Equations (6) and (7):

$$
p_1 = [T(x) \cdot T(y) \cdot T(z)]^{-1} \cdot E_A \cdot R(a) \cdot E_C \cdot R(c) \cdot p_2
\tag{6}
$$

$$
s_1 = R_{EA} \cdot R(a) \cdot R_{EC}(A_i) \cdot R(c) \cdot v_2
\tag{7}
$$

Based on the small error assumption, ignoring higher order terms, $E_A$ and $E_C$ in Equation (6) can be expressed as:

$$
E_A = T_{EA} \cdot R_{EA} \approx
\begin{bmatrix}
1 & -\varepsilon_{\gamma A} & \varepsilon_{\beta A} & \delta_{xA} \\
\varepsilon_{\gamma A} & 1 & -\varepsilon_{\alpha A} & \delta_{yA} \\
-\varepsilon_{\beta A} & \varepsilon_{\alpha A} & 1 & \delta_{zA} \\
0 & 0 & 0 & 1
\end{bmatrix},
$$

$$
E_C = T_{EC}(A_i) \cdot R_{EC}(A_i)
$$

$$
\approx
\begin{bmatrix}
1 & -\varepsilon_{\gamma C}(A_i) & \varepsilon_{\beta C}(A_i) & \delta_{xC1}(A_i) \\
\varepsilon_{\gamma C}(A_i) & 1 & -\varepsilon_{\alpha C}(A_i) & \delta_{yC1}(A_i) \\
-\varepsilon_{\beta C}(A_i) & \varepsilon_{\alpha C}(A_i) & 1 & \delta_{zC1}(A_i) \\
0 & 0 & 0 & 1
\end{bmatrix}.
$$

When the position $p_1 = (0, 0, 0, 1)^T$ of the starting tip point, the attitude of the starting tool, $s_1 = (0, 0, 1, 0)^T$ after the coordinate transformation to get the actual situation of the tool position expression $p' = (X', Y', Z', 1)^T$ and tool attitude expression as $v' = (I', J', K', 0)^T$ shown in Equations (8) and (9):

$$
p' = R(-c) \cdot [\ f_x(x, y, z, a) \quad f_y(x, y, z, a) \quad f_z(x, y, z, a) \quad 1\ ]^T
\tag{8}
$$

$$
v' = R(-c) \cdot [f_i(a)\ f_j(a)\ f_k(a)\ 1\ ]^T
\tag{9}
$$

Equations (8) and (9) unfolded to get Equation (10):

$$\begin{cases} X' = \cos c \cdot f_x(x,y,z,a) + \sin c \cdot f_y(x,y,z,a) \\ Y' = -\sin c \cdot f_x(x,y,z,a) + \cos c \cdot f_y(x,y,z,a) \\ Z' = f_z(x,y,z,a) \\ I' = \cos c \cdot f_i(a) + \sin c \cdot f_j(a) \\ J' = -\sin c \cdot f_i(a) + \cos c \cdot f_j(a) \\ K' = f_k(a) \end{cases} \tag{10}$$

where,

$$\begin{cases} f_x(x,y,z,a) = x + y \cdot \varepsilon_{\gamma A} - z \cdot \varepsilon_{\beta A} - \delta_{xA} + \varepsilon_{\gamma C} \cdot (y\cos a + z\sin a) \\ \qquad\qquad + \varepsilon_{\beta C} \cdot (y\sin a - z\cos a) - \delta_{xC1} \\ f_y(x,y,z,a) = -x \cdot \varepsilon_{\gamma C} + (y - x \cdot \varepsilon_{\gamma A} + z \cdot \varepsilon_{\alpha A} - \delta_{yA} + z \cdot \varepsilon_{\alpha C}) \cdot \cos a \\ \qquad\qquad + (z + x \cdot \varepsilon_{\beta A} - y \cdot \varepsilon_{\alpha A} - \delta_{zA} - y \cdot \varepsilon_{\alpha C}) \cdot \sin a - \delta_{yC1} \\ f_z(x,y,z,a) = x \cdot \varepsilon_{\beta C} + (x \cdot \varepsilon_{\gamma A} - z \cdot \varepsilon_{\alpha C} - y - z \cdot \varepsilon_{\alpha A} + \delta_{yA}) \cdot \sin a \\ \qquad\qquad + (x \cdot \varepsilon_{\beta A} - y \cdot \varepsilon_{\alpha C} - y \cdot \varepsilon_{\alpha A} + z - \delta_{zA}) \cdot \cos a - \delta_{zC1} \\ f_i(a) = -\varepsilon_{\beta A} + \sin a \cdot \varepsilon_{\gamma C} - \cos a \cdot \varepsilon_{\beta C} \\ f_j(a) = \cos a \cdot \varepsilon_{\alpha A} + \sin a + \cos a \cdot \varepsilon_{\alpha C} \\ f_k(a) = -\sin a \cdot \varepsilon_{\alpha C} - \sin a \cdot \varepsilon_{\alpha A} + \cos a \end{cases} \tag{11}$$

Equations (10) and (11) are mathematical expressions of the tip point position and tool attitude obtained by alignment coordinate transformation after the introduction of the rotation axis position error parameter. When all the rotation axis position error parameters tend to zero or are equal to zero, the derived error Equation (10) is equal to Equations (5) and (11), which represent the ideal tool position attitude.

The machine space geometry error is subtracted from the positional attitude of the tool and the theoretical position attitude in practice, as shown in Equation (12), and the spatial error field consisting of the points of space is analysed on the basis of Equation (12) as follows.

$$\begin{cases} E_p = p' - p \\ E_v = v' - v \end{cases} \tag{12}$$

$E_p$ represents the spatial position error of the tool nose point, and $E_v$ represents the spatial posture error of the tool. The spatial comprehensive error is $\Delta E$, as shown in Equation (13).

$$\Delta E = \sqrt{(X'-X)^2 + (Y'-Y)^2 + (Z'-Z)^2} \tag{13}$$

As the errors in the machining process of the machine tool comprise thermal errors, geometric errors, etc., this identification method based on the "S"-shaped test piece is mainly used to identify the geometric error of the machine tool.

## 3. Geometric Error Identification Based on the "S" Sample Family

### 3.1. Geometric Error Identification of the A-axis

There are six position-independent geometric errors in the A-axis of the dual-turntable five-axis CNC machining tool. These six errors can rapidly reflect the error distribution in the six degrees of freedom in space. In this section, a new identification method is proposed on the basis of an S-shaped specimen, and a mathematical model for the on-machine identification of the six position-independent geometric errors is established. Finally, the formulas to solve five positional errors are determined.

As shown in Figure 4, the established A-axis theoretical coordinate system is the actual coordinate system of $W_{iA}$, where the coordinate origin is at the theoretical intersection of the A-axes and C-axes. The S-shaped test piece is mounted on a rotating table, with three detection points on each of the two arcs, three detection points on the same plane, and the head perpendicular to the tangent plane, at the measuring point. The three measuring

points should be distributed as shown in Figure 4, where the solid points represent the view range visible points, and the hollow points represent the view range invisible points.

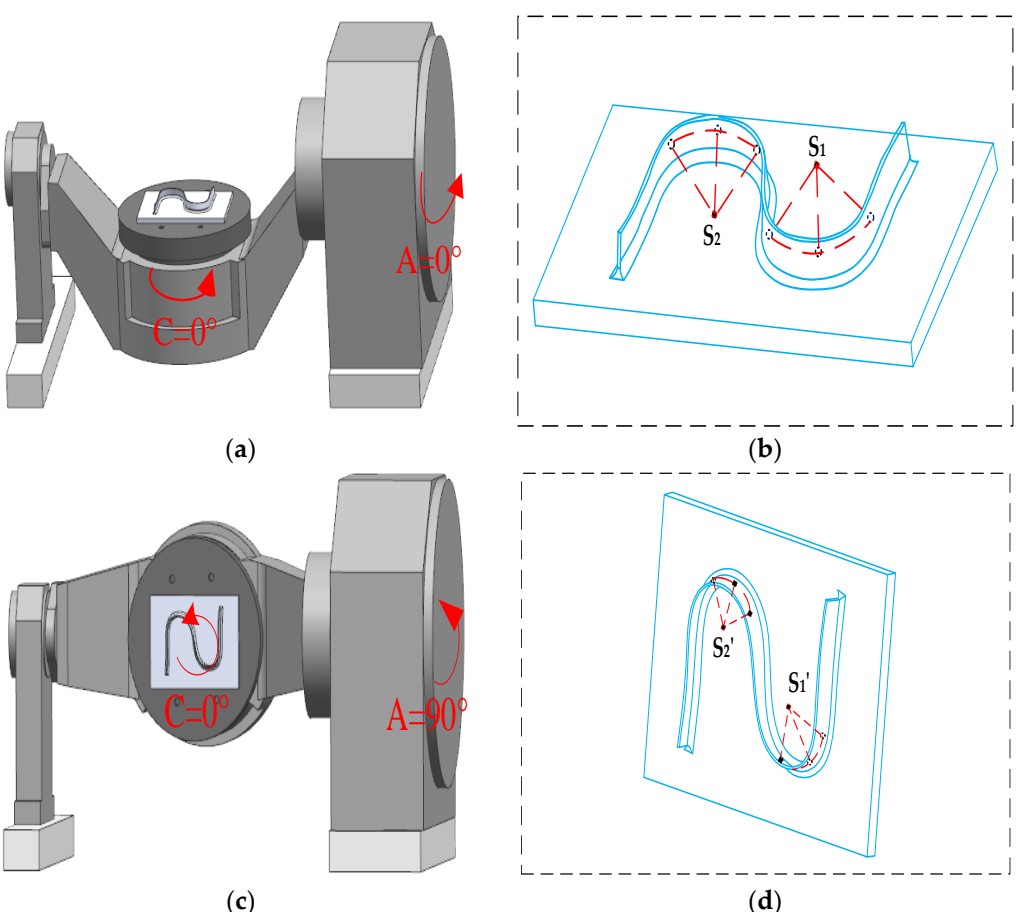

**Figure 4.** Schematic diagram of turntable position state when (**a**) a = 0° and (**c**) a = 90°, and marking diagram of measuring point position of S-shaped specimen at (**b**) a = 0° and (**d**) a = 90°.

$P1 = [xpypzp1]T$ denotes the centre point coordinates of the A-axis in the actual coordinate system when there is an error. $S_1$ and $S_1{'}$ are denote the centre point coordinates of the A-axis theoretical coordinate system when A = 0° and A = 90°, and $S_2$ is the same as $S_1$. The coordinates of the centre point of the circle can be obtained by matrix transformation from the measured coordinates of the centre of the circle, which is the transformation matrix between the machine tool coordinate system and the mathematical relationship between the $P_1$ and $S_1$ is as follows:

$$\begin{cases} S_1 = T_{AC} \cdot T_{EA} \cdot R_{EA} \cdot P_1 \\ S_1' = T_{AC} \cdot T_{EA} \cdot R_{EA} \cdot R(a) \cdot P_1 \end{cases} \tag{14}$$

$T_{AC} = \begin{bmatrix} 1 & 0 & 0 & m_x \\ 0 & 1 & 0 & m_y \\ 0 & 0 & 1 & m_z \\ 0 & 0 & 0 & 1 \end{bmatrix}$, $a = 90°$ further simplified to get the position error solution formula:

$$R(-a) \cdot R_{EA}{}^{-1} \cdot T_{EA}{}^{-1} \cdot T_{AC}{}^{-1} \cdot S_1' - R_{EA}{}^{-1} \cdot T_{EA}{}^{-1} \cdot T_{AC}{}^{-1} \cdot S_1 = 0 \tag{15}$$

This formula describes the relationship between the spatial position coordinates of the measuring point before and after the A-axis is rotated through 90°, and the geometric error elements of each position. The first set of Equation (16) used to solve the average positional

error of the A-axis is obtained by solving matrix Equation (15). The first set of equations is as follows:

$$\begin{cases} \Delta x_1 + \Delta y_1 \cdot \varepsilon_{\gamma A} - \Delta z_1 \cdot \varepsilon_{\beta A} = 0 \\ \varepsilon_{\gamma A} \cdot (z_1 - m_z + y_1' - m_y) = \delta_{yA} - \delta_{zA} + C_1 \\ \varepsilon_{\alpha A} \cdot (z_1' - m_z - y_1 + m_y) = \delta_{yA} + \delta_{zA} + C_2 \end{cases} \quad (16)$$

$\Delta x_1 = x_1' - x_1$, $\Delta y_1 = y_1' - y_1$, $\Delta z_1 = z_1' - z_1$, the second set of sphere coordinates of the "S" test piece are $S_2$ and $S_2'$. Using the same method, we get

$$\begin{cases} \Delta x_2 + \Delta y_2 \cdot \varepsilon_{\gamma A} - \Delta z_2 \cdot \varepsilon_{\beta A} = 0 \\ \varepsilon_{\gamma A} \cdot (z_2 - m_z + y_2' - m_y) = \delta_{yA} - \delta_{zA} + C_3 \\ \varepsilon_{\alpha A} \cdot (z_2' - m_z - y_2 + m_y) = \delta_{yA} + \delta_{zA} + C_4 \\ S_2 = \begin{bmatrix} x_2 & y_2 & z_2 & 1 \end{bmatrix}^T, S_2' = \begin{bmatrix} x_2' & y_2' & z_2' & 1 \end{bmatrix}^T \\ \Delta x_2 = x_2' - x_2, \Delta y_2 = y_2' - y_2, \Delta z_2 = z_2' - z_2 \end{cases} \quad (17)$$

After simultaneously solving Equations (16) and (17), the expression of the geometric error of the A-axis position can be obtained, as shown in Equation (18):

$$\begin{cases} \varepsilon_{\beta A} = (\Delta y_1 \Delta x_2 - \Delta x_1 \Delta y_2)/(\Delta y_1 \Delta z_2 - \Delta z_1 \Delta y_2) \\ \varepsilon_{\gamma A} = (\Delta z_1 \Delta x_2 - \Delta x_1 \Delta z_2)/(\Delta y_1 \Delta z_2 - \Delta z_1 \Delta y_2) \\ \varepsilon_{\alpha A} = (z_1 - z_2 + y_1' - y_2')/(C_1 - C_3) \\ \delta_{yA} = (z_1 + z_1' - 2m_z + \Delta y_1)(z_1 - z_2 + y_1' - y_2') \\ \qquad /(2C_1 - 2C_3) - (C_1 + C_2)/2 \\ \delta_{zA} = (y_1 + y_1' - 2m_y - \Delta z_1)(z_1 - z_2 + y_1' - y_2') \\ \qquad /(2C_3 - 2C_1) - (C_2 - C_1)/2 \end{cases} \quad (18)$$

The constant terms $C_1$, $C_2$, $C_3$ and $C_4$ in the formula are the error parameters obtained by the first step, and $\varepsilon_{\beta A}$, $\varepsilon_{\beta A}$ are the coordinate values of the two centre points, which are used to simplify the resulting positional error expression, as shown in Equation (19):

$$\begin{cases} C_1 = (x_1' - m_x) \cdot \varepsilon_{\beta A} + (x_1 - m_x) \cdot \varepsilon_{\gamma A} - y_1 + z_1' + m_y - m_z \\ C_2 = (m_x - x_1) \cdot \varepsilon_{\beta A} + (x_1' - m_x) \cdot \varepsilon_{\gamma A} - y_1' - z_1 + m_y + m_z \\ C_3 = (x_2' - m_x) \cdot \varepsilon_{\beta A} + (x_2 - m_x) \cdot \varepsilon_{\gamma A} - y_2 + z_2' + m_y - m_z \\ C_4 = (m_x - x_2) \cdot \varepsilon_{\beta A} + (x_2' - m_x) \cdot \varepsilon_{\gamma A} - y_2' - z_2 + m_y + m_z \end{cases} \quad (19)$$

The five positional error elements obtained by the solution are substituted into Equation (15) to obtain the position error $\delta_{xA}$.

### 3.2. Geometric Error Identification of Axis C

The process of identifying C-axis positioning errors is similar to that of the A-axis, however, it is necessary to solve various C-axis positional errors at different A-axis rotation angles to analyse its rules. These can be divided into three angles: $A_1 = 0°$, $A_2 = 45°$, $A_3 = 90°$. The six average positional errors of the A-axis were identified in the previous section. Since the C-axis is connected with the A-axis, the positional errors of the A-axis need to be added into the C-axis positional errors identification process.

Figure 5 shows the detection process of C-axis positional errors. $W_{iC}$ is the theoretical coordinate system of the C-axis corresponding to that of axis A which is $W_{iA}$, and the actual coordinate system of the C-axis error is $W_{rC}$. It is assumed that the two central points ($O = \begin{bmatrix} x & y & z & 1 \end{bmatrix}^T, O' = \begin{bmatrix} x' & y' & z' & 1 \end{bmatrix}^T$) are obtained by probe measurement via a machine tool coordinate system (MCS). The coordinates of the central points $O$ and $O'$ can be transformed by the transformation matrix $T_{AC}$, and can be obtained as S in the theoretical coordinate system $W_{iC}$. The coordinate transformation relationship between them is: $O = T_{AC}^{-1} \cdot S$.

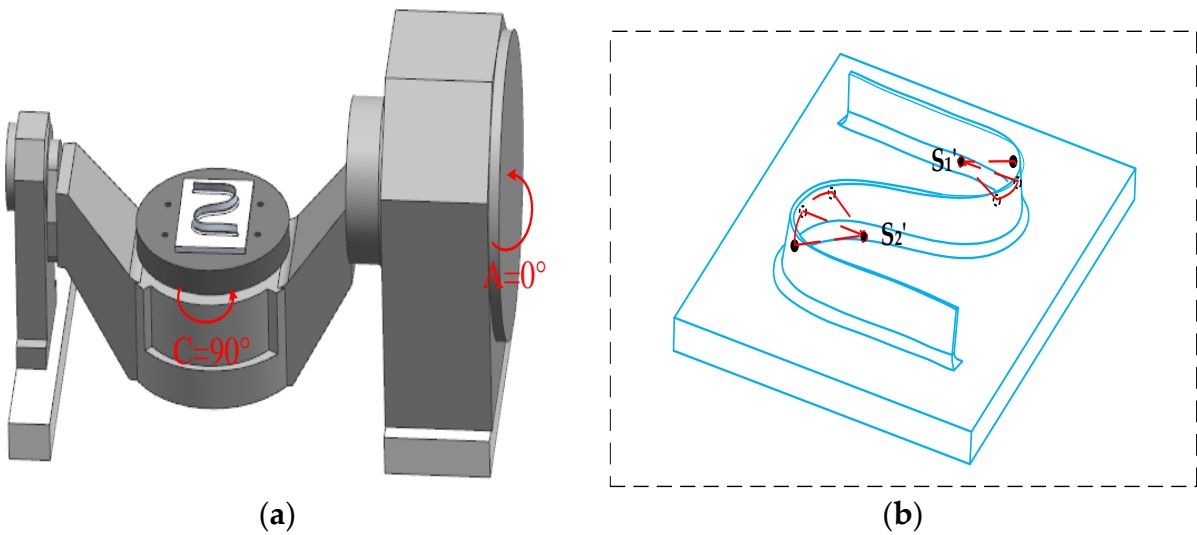

**(a)**                           **(b)**

**Figure 5.** (**a**) Schematic diagram of turntable position when C = 90°, and (**b**) location marking diagram of s-shaped specimen sampling point when C = 90°.

According to the coordinate transformation relationship between coordinate $P_1$ of the central point in $W_{rC}$ and the coordinates $S_1$, $S_1'$, $S_2$, and $S_2'$ of the two groups of the central points with errors, the following Equation (20) can be obtained:

$$\begin{cases} S_1 = T_{EA} \cdot R_{EA} \cdot R(a) \cdot T_{EC} \cdot R_{EC} \cdot P_1 \\ S_1' = T_{EA} \cdot R_{EA} \cdot R(a) \cdot T_{EC} \cdot R_{EC} \cdot R(c) \cdot P_1 \end{cases} \tag{20}$$

The two equations can be expanded and subtracted to obtain the C-axis positional error solution formula:

$$R(-c) \cdot R_{EC}^{-1} \cdot T_{EC}^{-1} \cdot R(-a) \cdot R_{EA}^{-1} \cdot T_{EA}^{-1} \cdot T_{AC}^{-1} \cdot S_1' - R_{EC}^{-1} \cdot T_{EC}^{-1} \cdot R(-a) \cdot R_{EA}^{-1} \cdot T_{EA}^{-1} \cdot T_{AC}^{-1} \cdot S_1 = 0 \tag{21}$$

Two C-axis skew errors are obtained following matrix operation, which are $\varepsilon_{\alpha C}$ and $\varepsilon_{\beta C}$, as shown in Equations (22) and (23), where the positional error of axis A is substituted as a known constant.

$$\varepsilon_{\alpha C} = -\varepsilon_{\alpha A} + (\Delta x_1 \Delta z_2 - \Delta z_1 \Delta x_2)/(\Delta x_1 \Delta y_2 - \Delta y_1 \Delta x_2) \tag{22}$$

$$\varepsilon_{\beta C} = -\varepsilon_{\beta A} + (\Delta y_1 \Delta z_2 - \Delta z_1 \Delta y_2)/(\Delta x_1 \Delta y_2 - \Delta y_1 \Delta x_2) \tag{23}$$

$$\varepsilon_{\gamma C} = \frac{(z_1 + z_1' - z_2 - z_2') \cdot \varepsilon_{\beta C} + (\Delta z_1 - \Delta z_2) \cdot \varepsilon_{\alpha C} + C_5 - C_6}{(y_1 + y_1' + \Delta x_1 - y_2 - y_2' - \Delta x_2)} \tag{24}$$

It can be seen from Equation (24) that the tilt error of the C-axis rotation around the Z-axis is related to the other two C-axis tilt errors. The two key line errors $\delta_{xC1}$ and $\delta_{yC1}$ of the C-axis can be obtained from Equations (25) and (26). The results are as follows:

$$\delta_{xC1} = \left[(y_1 + y_1' + \Delta x_1) \cdot \varepsilon_{\gamma C} - (z_1 + z_1') \cdot \varepsilon_{\beta C} - \Delta z_1 \cdot \varepsilon_{\alpha C} - C_5\right]/2 \tag{25}$$

$$\delta_{yC1} = \left[(y_1' - y_1 - x_1' - x_1) \cdot \varepsilon_{\gamma C} - \Delta z_1 \cdot \varepsilon_{\beta C} + \Delta z_1 \cdot \varepsilon_{\alpha C} + C_7\right]/2 \tag{26}$$

$C_5$, $C_6$ and $C_7$ are shown in Equation (27):

$$\begin{cases} C_5 = \Delta z_1 \cdot \varepsilon_{\alpha A} + (z_1 + z_1') \cdot \varepsilon_{\beta A} - (y_1 + y_1' + \Delta x_1) \cdot \varepsilon_{\gamma A} + 2\delta_{xA} - x_1 - x_1' + \Delta y_1 \\ C_6 = \Delta z_2 \cdot \varepsilon_{\alpha A} + (z_2 + z_2') \cdot \varepsilon_{\beta A} - (y_2 + y_2' + \Delta x_2) \cdot \varepsilon_{\gamma A} + 2\delta_{xA} - x_2 - x_2' + \Delta y_2 \\ C_7 = (z_1 + z_1') \cdot \varepsilon_{\alpha A} - \Delta z_1 \cdot \varepsilon_{\beta A} - (x_1 + x_1' - \Delta y_1) \cdot \varepsilon_{\gamma A} - 2\delta_{yA} + \Delta x_1 + y_2' + y_1 \end{cases} \tag{27}$$

Using the S-shaped specimen family, the values of the measuring points can be obtained by the identification methods described in Sections 3.1 and 3.2, and the actual central points can then be determined by the three points forming the centre of the circle. The geometric error value of the machine tool can be identified by entering the value of the central points into the above geometric error identification model.

## 4. In-Machine Measurement of the Sampling Point Error of S-Shaped Samples

### 4.1. Mathematical Model of the S-Shaped Sample Family

The shape of the S-shaped specimen is unique and has the characteristics of rich curvature changes and drastic continuity changes. A straight grain surface can be obtained by sweeping the upper and lower two uniform B-spline curves three times as the sweep track, and a straight line is swept according to this track. The upper and lower wires of the S-shaped specimens were constructed using a third-order B-spline curve with second-order continuous conduction. The mathematical definition of the B-spline curve is given as:

$$Q(u) = \sum_{i=0}^{n} d_i N_{i,k}(u) \tag{28}$$

where $d_i$,—the control vertex of a curve; $N_{i,k}(u)$—the basis function of the $k$-degree spline curve; $k$—the number of B splines; and $u$—the interpolation variables of spline curves, $u \in [0, 1]$.

The terminal fixing condition of the S-shaped specimen is composed of quadruple nodes, with $u_0 = u_1 = u_2 = u_3 = 0, u_{n+1} = u_{n+2} = u_{n+3} = u_{n+4} = 1$ and $k = 3$. The B-spline curve was established via the De Boer recursion method, and the curve was expressed in matrix form as follows:

$$Q_i(u) = \frac{1}{6}[u^3, u^2, u, 1] = \begin{bmatrix} -1 & 3 & -3 & 1 \\ 3 & -6 & 3 & 0 \\ -3 & 0 & 3 & 0 \\ 1 & 4 & 1 & 0 \end{bmatrix} \begin{bmatrix} d_i \\ d_{i+1} \\ d_{i+2} \\ d_{i+3} \end{bmatrix} \tag{29}$$

The upper and lower wire models of the S-shaped specimen were established based on Equation (29), as shown in Figure 6a, and the S-shaped specimen measuring points' distribution is shown in Figure 6b.

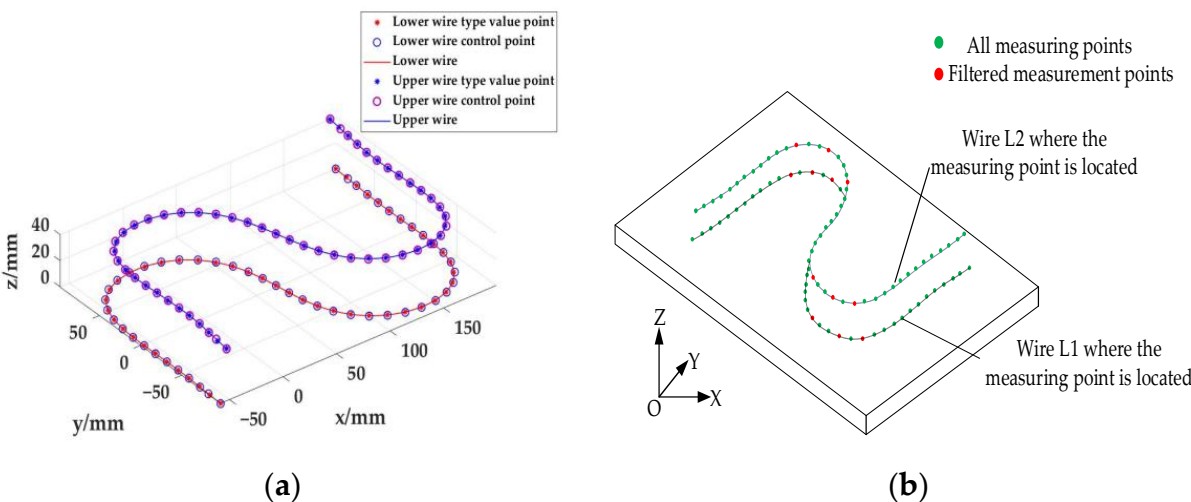

**(a)**          **(b)**

**Figure 6.** S-shaped specimen model and distribution diagram of measuring points for (**a**) Upper and lower wire models of S-shaped specimens and (**b**) Sampling point distribution diagram of S-shaped specimens.

*4.2. In-Machine Measurement of Sampling Point Error Based on the S-Shaped Specimen*

The S-shaped specimen used in the machine measurement was processed using a ball end milling cutter on a higher precision machining tool, with a ball end milling cutter specification of R2.0-8L-4D-50L. The material of the S-shaped specimen is aviation aluminium alloy, and the processing standard is ISO10791-7:20 [20]. In the experiment, sampling points were devised according to the 3-D geometric model of the S-shaped samples. The sampling points were in accordance with the reference measurement points given in ISO 10791-7:20. A total of fifty coordinate points were taken from the upper and lower wires of the S-shaped specimen, as shown in Figure 6. The probe was then used for an in-machine measurement of the S-shaped sample, as shown in Figure 7.

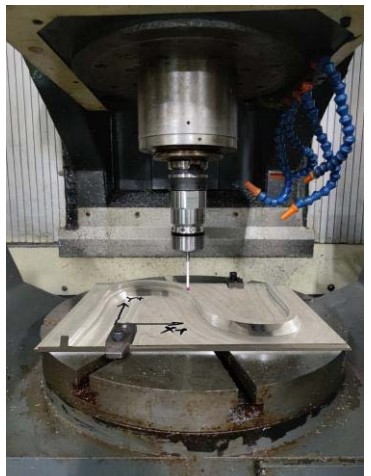 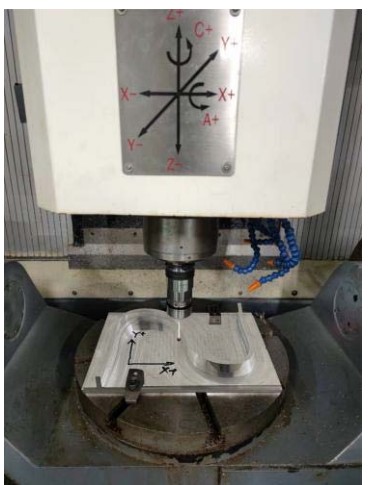 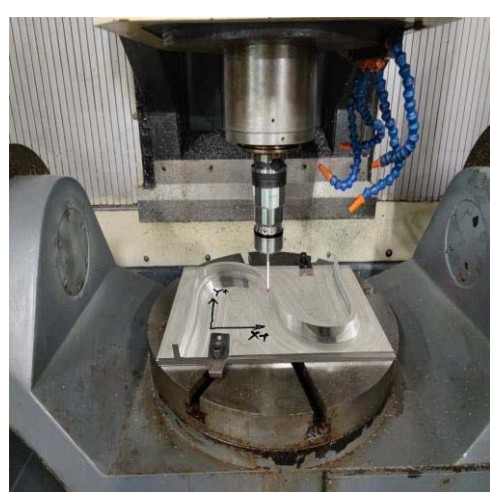

**Figure 7.** In the field of machine measurement.

Based on in-machine measurement, fifty coordinate measuring points on the upper and lower wires of the S-shaped specimen were measured when the angle of axis A was 0°, 30°, 60°, and 90°, and the angle of axis C was 0°, 30°, 60° and 90°, respectively. The specific measurement process is shown in Figure 8.

After measuring the given coordinate points of the S-shaped specimen, the coordinate values were compared to the coordinate values of the reference measurement points given in the relevant ISO 10791-7:2020 standards. When the A-axis and C-axis equal 0°, S-shaped specimens with curved surface inclination angles of 0° and 15° (abbreviated as "0° S-shaped specimen" and "the error measurement value of 15° S-shaped specimen") occur, as shown in Figure 9. According to the variation law of the error in each direction and the comprehensive error, the error of the two S specimens in the constant curvature area is small; the error in the variable curvature part starts to increase rapidly, and the distortion-free section reaches a relatively stable level in the middle part of the specimen. That is, the distorted part is changed, and the error in each direction reaches a maximum value. Due to the drastic change in the test piece curvature, the error in the X and Y directions of the 0° S-shaped test piece also changes drastically, along with its direction. For the spatial comprehensive error, the overall trend of the two S-shaped specimens shows that from the first measuring point to the 50th measuring point, the comprehensive error initially increases and then decreases, and the error reaches its maximum at the centre of the specimen.

The curvature changes of S-shaped specimens can be classified into constant curvature, variable curvature, no distortion, and variable distortion. The coordinates of 50 measuring points are distributed in each region. Tables 2 and 3 respectively show S-shaped specimens with curved surface inclination angles of 0° and 15°.

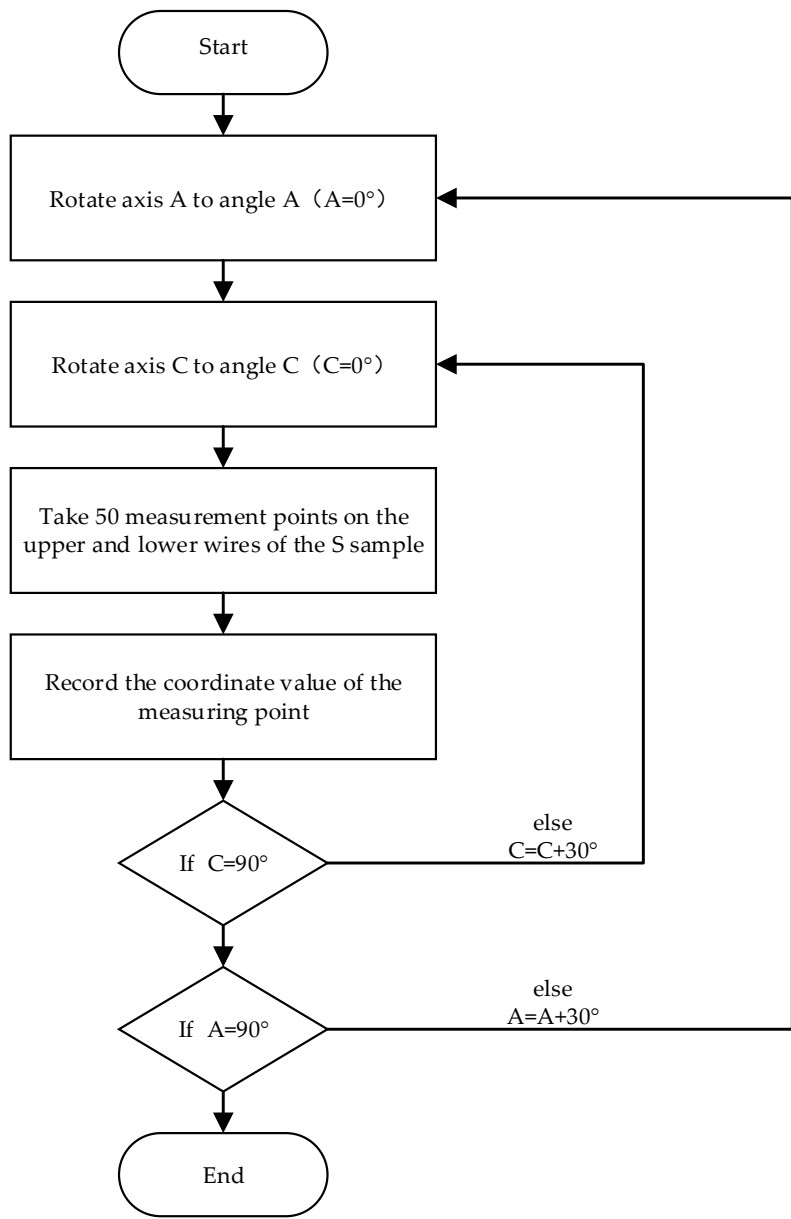

**Figure 8.** Measurement flow chart.

It can be seen from Tables 2 and 3 that the maximum error value of the variable distortions of the S-shaped curve is the largest between the 23rd and 30th measuring points. In the constant curvature part of the curve, the maximum error value is the smallest, as is the minimum error value. According to Table 3, the maximum error value reaches its peak value in the section of the curve without distortion, and the minimum is in the constant curvature section. At the same time, the minimum error value reaches its maximum in the untwisted part of the curve and its minimum in the constant curvature section. Comprehensive analysis shows that the maximum and minimum error values of S-shaped specimen are larger in the variable curvature, variable distortion, and no distortion sections, and smaller in the constant curvature section.

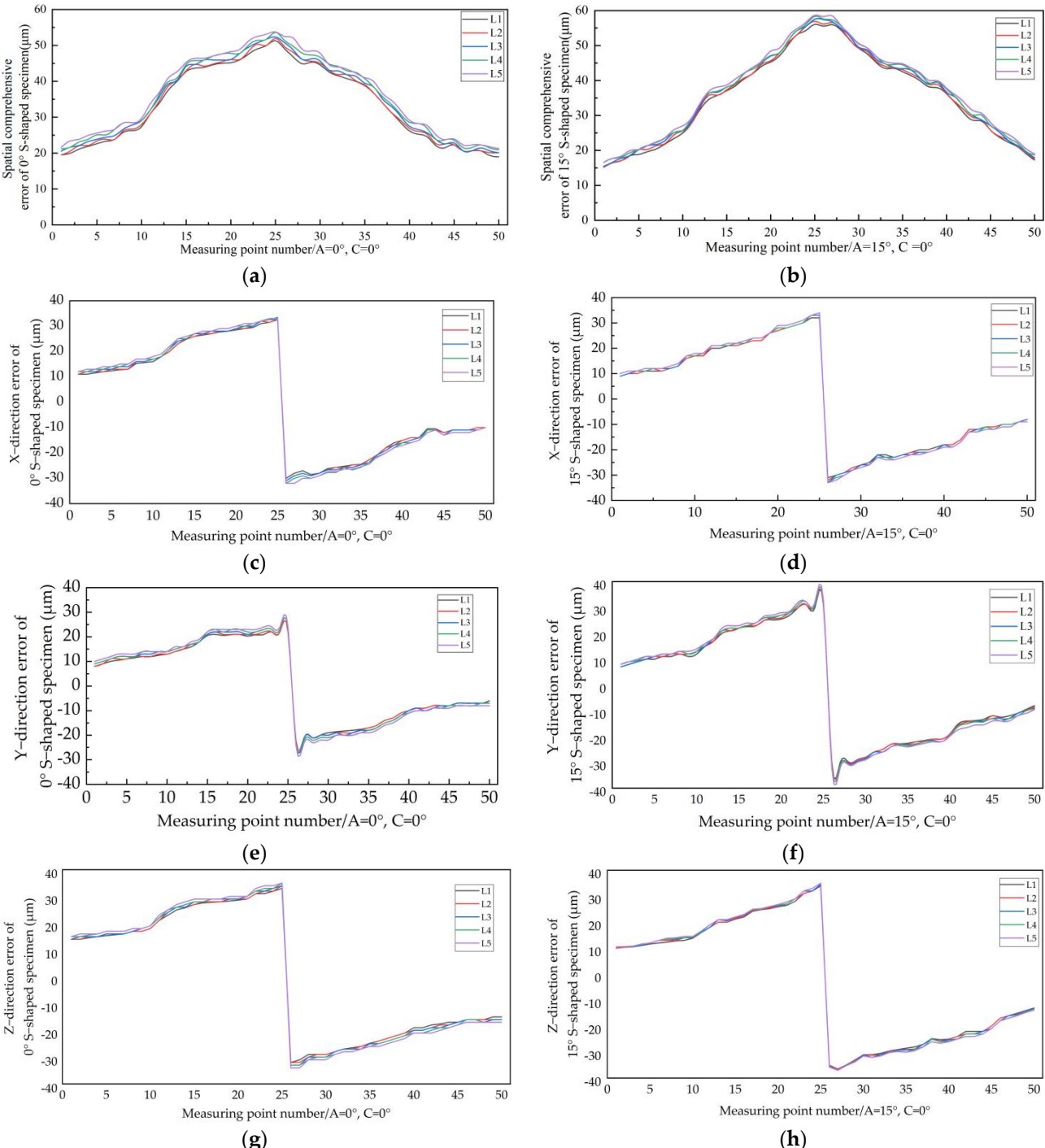

**Figure 9.** Different curved surface inclination angles for comprehensive error of S-shaped specimen at (**a**) 0° and (**b**) 15°, and X-direction error of S-shaped specimen at (**c**) 0° and (**d**) 15°, and Y direction error of S-shaped specimen at (**e**) 0° and (**f**) 15°, and Z direction error of S-shaped specimen at (**g**) 0° and (**h**) 15°.

**Table 2.** Comprehensive error data distribution of specimens with 0° inclined plane angle.

| Curvature Change of Specimen | Constant Curvature | Variable Curvature | No Distortion | Distorted | No Distortion | Variable Curvature | Constant Curvature |
|---|---|---|---|---|---|---|---|
| Measurement point | 1~10① | 11~14② | 15~22③ | 23~30④ | 31~36⑤ | 37~41⑥ | 42~50⑦ |
| Maximum error (μm) | 30 | 43 | 51 | 54 | 45 | 37 | 28 |
| Minimum error (μm) | 20 | 31 | 43 | 45 | 37 | 25 | 19 |
| Mean error (μm) | 25 | 37 | 46 | 49 | 41 | 31 | 22 |

**Table 3.** Comprehensive error data distribution of specimens with 15° inclined plane angle.

| Curvature Change of Specimen | Constant Curvature | Variable Curvature | No Distortion | Distorted | No Distortion | Variable Curvature | Constant Curvature |
|---|---|---|---|---|---|---|---|
| Measurement point | 1~10① | 11~14② | 15~22③ | 23~30④ | 31~36⑤ | 37~41⑥ | 42~50⑦ |
| Maximum error (μm) | 27 | 38 | 53 | 59 | 49 | 42 | 34 |
| Minimum error (μm) | 15 | 28 | 37 | 49 | 41 | 34 | 17 |
| Mean error (μm) | 21 | 34 | 44 | 55 | 45 | 38 | 25 |

The first two centres on the upper measuring wire can be determined according to the S-shaped specimen comprehensive error analysis, as shown in Table 2. The measuring point closest to the mean error value is taken in the constant curvature ①, variable curvature ②, and no distortion ③ sections, respectively, and the specific coordinate value of the centre of the circle is calculated by using the three-point Formula (30). In the without distortion ⑤, variable curvature ⑥, and constant curvature ⑦ sections, a measuring point nearest to the mean error value is taken, and the specific coordinate value of the centre of the circle is calculated. Then, the two centres on the lower measuring wire are determined. Because the curvature direction of the variable twist area ④ changes and the error changes drastically, the coordinate value of the center of the circle is not easy to determine. Therefore, the measurement point closest to the average error value is not se-lected in the variable twist area ④. The process of determining the centre of the circle by three points is as follows: taking the three points on the plane as $(x_1, y_1)$, $(x_2, y_2)$, and $(x_3, y_3)$, assuming that $a = x_1 - x_2$, $b = y_1 - y_2$, $c = x_1 - x_3$, $d = y_1 - y_3$, $e = \frac{(x_1^2 - x_2^2) - (y_2^2 - y_1^2)}{2}$, $f = \frac{(x_1^2 - x_3^2) - (y_3^2 - y_1^2)}{2}$, The coordinates of the centre of the circle can be obtained from the parameters set above:

$$\begin{cases} x_0 = -\frac{de - bf}{bc - ad} \\ y_0 = -\frac{af - ce}{bc - ad} \end{cases} \tag{30}$$

According to Formula (30), four centres are obtained, and the two with the largest spatial distance are selected. The specific determination process of the centre coordinates of the upper and lower wires is shown in Figure 10. According to the identification method described in Section 3, these data points are used to form the centre of the circle, and the coordinates obtained are substituted into Equations (18), (19) and (22)–(27). Twelve geometric error elements such as A-axis movement errors $\delta_{xA}$, $\delta_{yA}$, and $\delta_{zA}$, rotation errors $\varepsilon_{\alpha A}$, $\varepsilon_{\beta A}$, and $\varepsilon_{\gamma A}$, as well as C-axis movement errors $\delta_{xC}$, $\delta_{yC}$, and $\delta_{zC}$, and rotation errors $\varepsilon_{\alpha C}$, $\varepsilon_{\beta C}$, and $\varepsilon_{\gamma C}$ can be obtained.

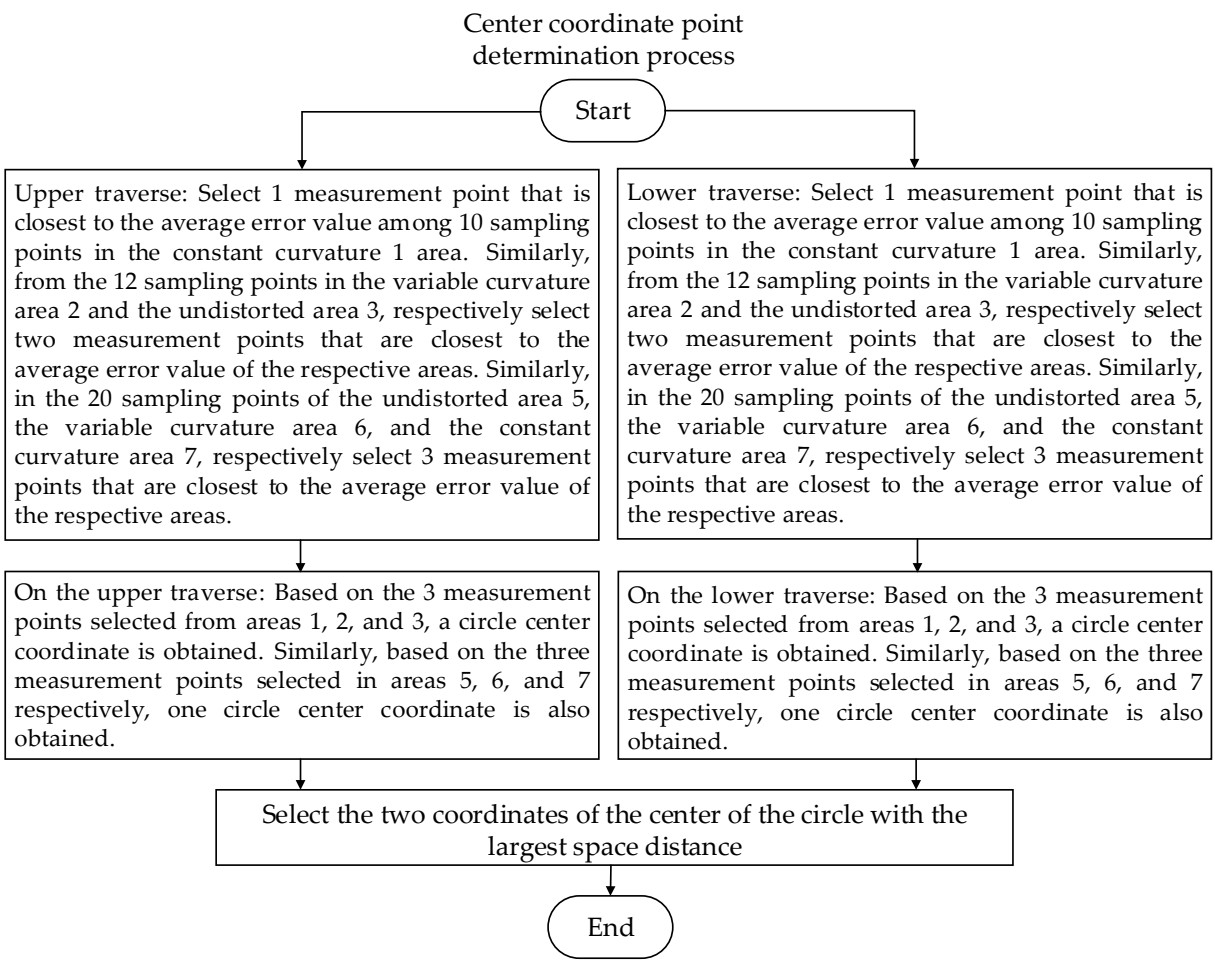

**Figure 10.** Flow chart for determining the centre coordinates of the upper and lower wires.

## 5. Establishment and Analysis of the Machining Space Comprehensive Error Field

Using the geometric error model represented by Equations (10)–(12), the geometric error field of the working space of the five-axis machine tool was established with five A-axis rotation angles, A = 0°, A = 30°, A = 45°, A = 60°, and A = 90° to obtain the variation rule of the spatial error field during the A-axis swing. Based on the comprehensive error data in Tables 2 and 3, the comprehensive error field distribution can be plotted, as shown in Figure 11.

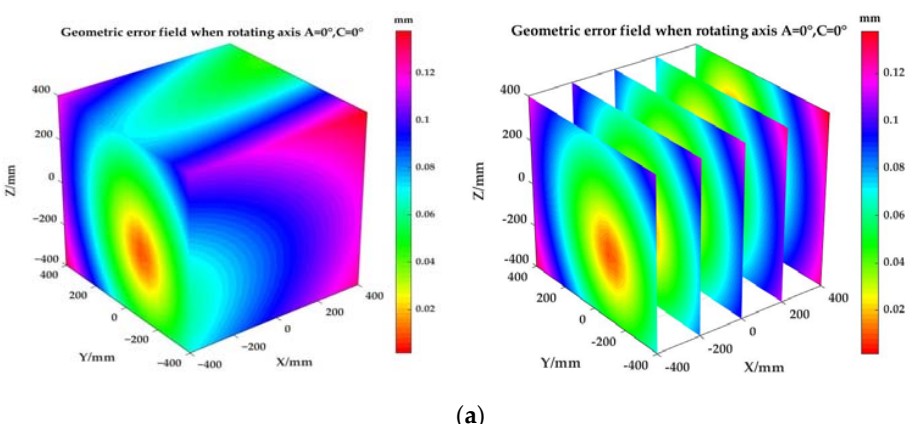

(a)

**Figure 11.** *Cont.*

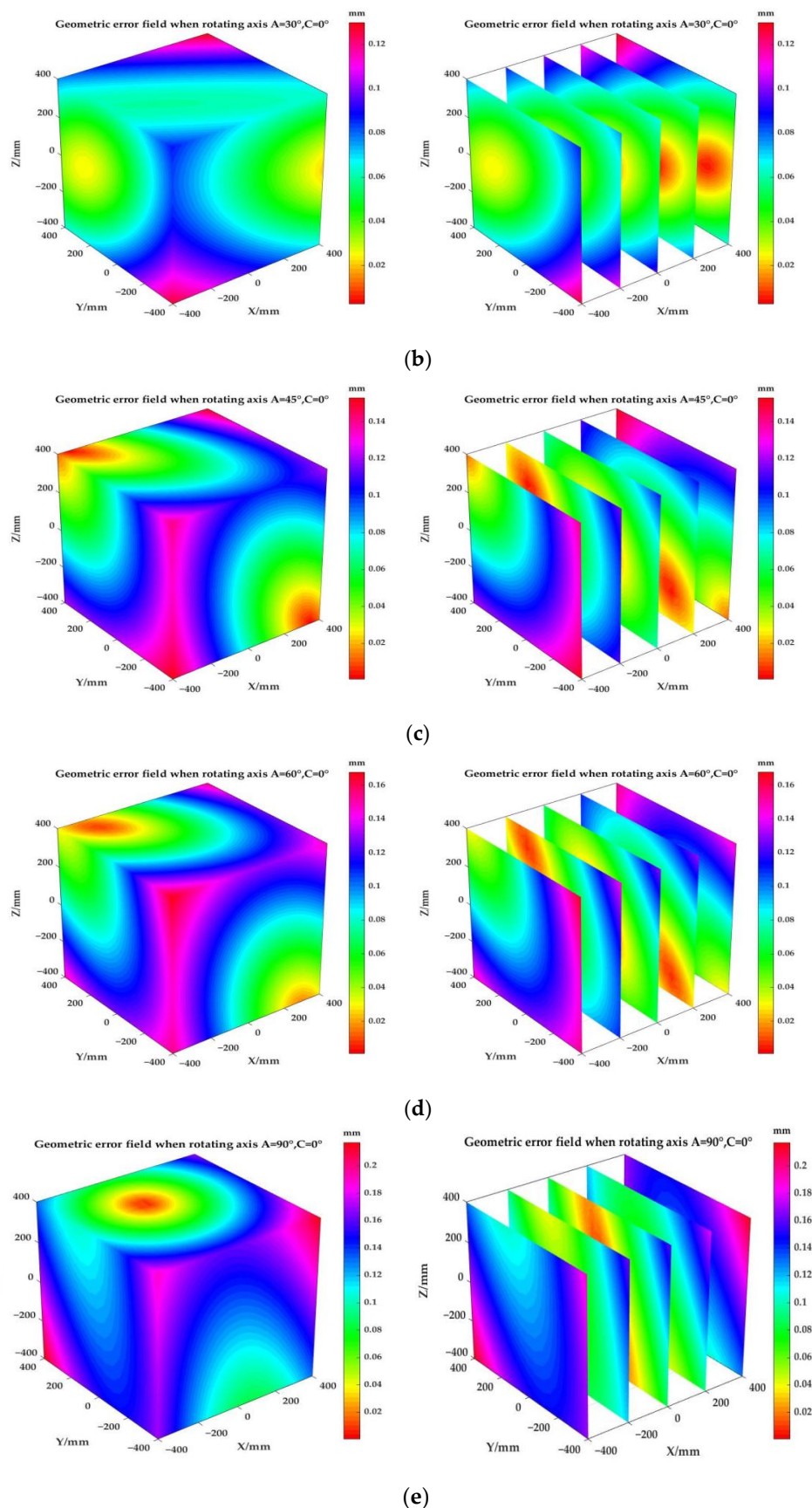

**Figure 11.** The distribution of the comprehensive error field at all angles of the A-axis at $0°$ C axis for (**a**) A = $0°$, (**b**) A = $30°$, (**c**) A = $45°$, (**d**) A = $60°$ and (**e**) A = $90°$.

By analysing each error field, it can be seen that when A = 0°, as shown in Figure 11a, the rotation axis spatial comprehensive error of the five-axis CNC machine tool increases from the centre outwards. When the workspace is located near the origin of the ideal coordinate system on axes A and C, the synthetic errors are less than 4 µm at all angles. When the workspace expands outwards, the synthesis error gradually increases, reaching a maximum of about 122 µm at the edge of the workspace. In Figure 12b,c, when the rotation angle of the C-axis is less than 90° of the maximum comprehensive error range the C-axis rotation angle is increased. In Figure 12d, when the C-axis rotation angle reaches 90°, the comprehensive error maximum value reaches 234 µm, and the minimum error also changes with the difference of the C-axis angle. It can be seen that the error caused by large rotation angles is also large. Therefore, in the cutting process, we should try to avoid excessive A-axis angle rotation, which can effectively reduce the influence of machine tool geometric error on machining accuracy.

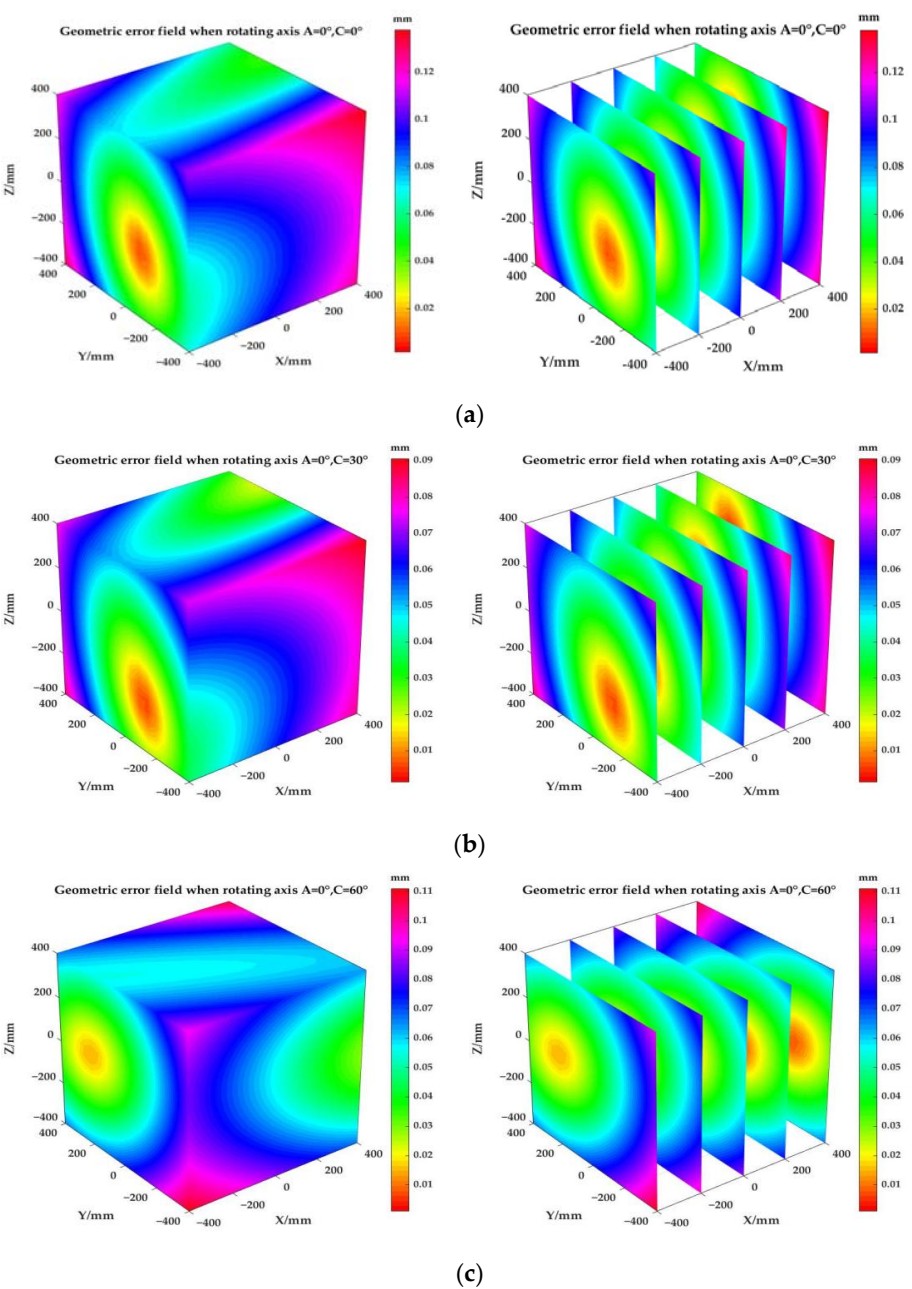

**Figure 12.** *Cont.*

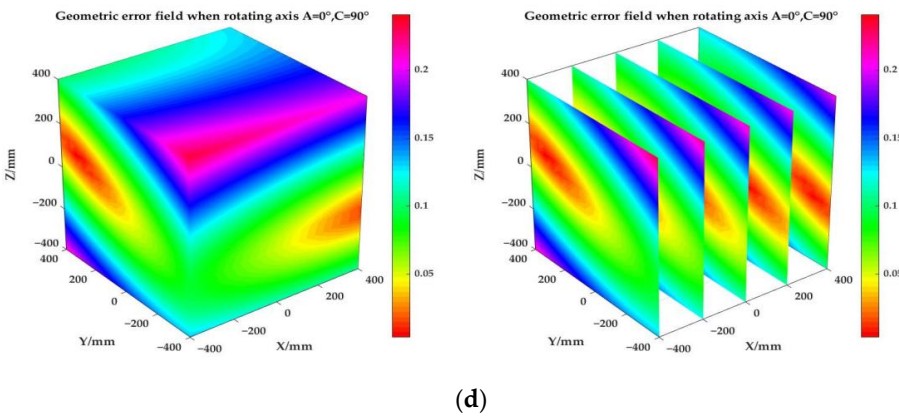

**(d)**

**Figure 12.** The distribution of the comprehensive error field at all angles of the C-axis at 0° A axis for (**a**) C = 0°, (**b**) C = 30°, (**c**) C = 60° and (**d**) C = 90°.

The coordinate values of data points measured in the machine are substituted into the geometrical error model of the rotation axis to calculate the comprehensive error values of axes A and C of the machine tool rotation via Formula (12). Tables 4 and 5 show the comprehensive error values of axes C and A at different angles. It can be seen from Table 4 that when the C-axis is 0° and the rotation angle is unchanged, the maximum comprehensive error value of the machine tool increases correspondingly with the increase of the A-axis rotation angle, and the error value reaches its maximum when the A-axis rotation angle is 90°. Meanwhile, it can be seen from Table 4 that when the A-axis is 90°, the minimum comprehensive error of the machine tool reaches its minimum. According to Table 5, when the angle of the A-axis is 0° and the rotation angle remains unchanged, the maximum value of the comprehensive error of the machine tool also demonstrates an increasing trend. When the angle of the C-axis is 90°, the maximum value of the comprehensive error reaches its limit. Similarly, when the rotation angle of the A-axis is unchanged, the minimum comprehensive error of the machine tool reaches its minimum when the C-axis angle is 30°.

**Table 4.** The comprehensive error of the A-axis at each angle states when the C-axis = 0°.

| Compre-Hensive Error | A-Axis Angle 0° | 30° | 60° | 90° |
|---|---|---|---|---|
| $\Delta E_{\min}(mm)$ | 0.004 | 0.008 | 0.003 | 0.003 |
| $\Delta E_{\max}(mm)$ | 0.122 | 0.127 | 0.167 | 0.217 |

**Table 5.** The comprehensive error of the C-axis at each angle states when the A-axis = 0°.

| Compre-Hensive Error | C-Axis Angle 0° | 30° | 60° | 90° |
|---|---|---|---|---|
| $\Delta E_{\min}(mm)$ | 0.004 | 0.003 | 0.004 | 0.004 |
| $\Delta E_{\max}(mm)$ | 0.122 | 0.084 | 0.111 | 0.234 |

## 6. Conclusions

(1) Based on the "S"-shaped specimen family, the error mapping relationship between the machining process system and the workpiece is analyzed, and the 12 geometric errors of the two rotation axes are identified by the double-center coordinate values obtained from the sampling points measured on the specimen. A comprehensive error field prediction method for five-axis machine tool machining space based on the "S"-shaped specimen family is proposed. Compared with the position error identification

method based on double standard spheres, the geometric error identified by this method is about 90% consistent.

(2) The five-axis CNC machine rotation axis error field was plotted. As can be seen from the comprehensive error field prediction image, when A = 90° the comprehensive error is smallest at the centre of the processing space, at 0.003 mm, and the internal-to-external error is gradually increasing. The comprehensive error increases with the increase in the rotation angle of the A-axis; therefore, excessive angle rotation of the A-axis should be avoided during processing. The same predictive image by the comprehensive error field can be seen in the A-axis angle. To maintain the constant C-axis angle from 0° to 30°, the angle gradually increased, the machine's processing comprehensive error gradually increased, therefore excessive angle rotation of the C-axis during processing should also be avoided.

(3) This paper proposes using S-shaped test pieces manufactured by high-precision machine tool processing as a sample of machine geometry error detection, which is cost-effective. However, considering that there are not only geometric errors in the processing of "S" test pieces, but also thermal errors and machine vibration, these factors may make the measurement of geometric errors inaccurate. Therefore, it is necessary to further study the separation and identification of machine tool processing errors in the future.

**Author Contributions:** Conceptualization, S.W.; methodology, Z.F.; software, F.Q.; validation, Z.D.; formal analysis, Z.D.; resources, F.Q.; data curation, Z.D.; writing—original draft preparation, Z.D.; writing—review and editing, S.W.; project administration, S.W. All authors have read and agreed to the published version of the manuscript.

**Funding:** This research work was supported by the National Natural Science Foundation of China (No. 51720105009).

**Institutional Review Board Statement:** Not applicable.

**Informed Consent Statement:** Not applicable.

**Data Availability Statement:** Not applicable.

**Conflicts of Interest:** Conflicts of interest on behalf of all authors, the corresponding author states that there is no conflict of interest.

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
