# Peer review of "Prediction of the Comprehensive Error Field in the Machining Space of the Five-Axis Machine Tool Based on the “S”-Shaped Specimen Family"

_machines, doi:10.3390/machines10050408_

Round 1

Reviewer 1 Report

The article presents an analysis of the assessment of the expected machining errors while working on a five-axis CNC machine tool. the subjected to analysis includes processing and evaluation of the shape on a coordinate measuring machine. The work is interesting from the point of view of the practical implementation of the research. 
The article requires some corrections and some areas of work to be improved and more detailed explained to readres of some details.
The abstract is a little bit confuse and missis some information like more results and conclusions, I suggest to authors follow these rules:
- One or two sentences on BACKGROUND
- Two or three sentences on METHODS
- Less than two sentences on RESULTS
- One sentence on CONCLUSIONS
The introduction chapter is preceded by the digit "0", so there are many unnecessary inclusions in the content - unexpected numbers, maybe literature footnotes, should be in parentheses. The author has not shown due diligence in the preparation of this chapter. 
Unfortunately, Figure 1 does not show the structure of the machine tool. The semi 3D model raises some doubts. I am asking for an isometric presentation of the model with explanations of the machine components. Drawing presentation quality to be improved.
Coordinates topolgy should be better presented - figure 2. It seems to be a better solution to relate machine components directly to coordinate systems and to define proper movements for rotational and linear.
Based on the presented equations, it can be concluded that the error in the geometric structure of the machine is subtracted from the actual position of the tool, the error of which may be influenced by a number of factors, for example thermal and, consequently, mechanical. Can the size of these errors have an impact on the presented assessment? 
Captions under figures require correction, description (a), (b) under the figure should be not in the graphics area.
Figure 5, illegible axle rotation markings.
Line 325,338,  ISO Standard - Literature Reference Required
Figure 9 - charts, no units and markings on the axes, to be completed. The quality of the charts is very poor 
Graphing the results is difficult for the reader. It seems necessary to present the results on a line graph for various functions of the table torsion angle for the magnitude of errors for the X, Y, Z axes. It will then be easier to analyze it visually. 
The author consistently in the text refers to the footnotes of the literature to only the author's name - no numeric tag - line 479, this must be standardized and corrected throughout the work.

Author Response

Responses to the comments for “ Prediction of Comprehensive Error Field in Machining Space of Five-axis Machine Tool Based on S-shaped Specimen Family ”(machines-1622515)

Dear editors and reviewers:

Many thanks for the insightful comments and suggestions of the reviewers. Considering the reviewers' comments and suggestions, I have made in responses to the reviewers' questions and suggestions on an item by item basis. The amendments are made revisions to the manuscript using Track Changes. The following is the responses to the comments for “Prediction of Comprehensive Error Field in Machining Space of Five-axis Machine Tool Based on "S"-shaped Specimen Family”.

We appreciate for Editors/Reviewers’ warm work earnestly, and hope that the correction will meet with approval.

With best wishes

Yours sincerely

Shi Wu

Reviewer 2 Report

As a cost-effective scheme for detecting machine geometry errors, this paper proposes the use of S-shaped specimens fabricated with high-precision machine tools.
I have just a few suggestions that may improve the clarity and quality of this paper:
1. The novelty of the paper is assessed in the Conclusions section, and the paragraph " Li used the standard sphere measurement method to obtain 10 average positional errors of the two rotating axes by inputting them into the coordinate system. Compared with the traditional measurement method, the accuracy of this method is about 91.8%, but the cost is lower, and the measurement time is shorter" should be moved to the end of the Introduction. Cite Li's research when mentioning it.

2. Improvements need to be made to Figures 1 and 3. Avoid using light blue color for notes.

3. An overall flowchart of the research would be useful. 

4. It is important to rephrase the conclusions in a concise manner.

Author Response

Responses to the comments for “ Prediction of Comprehensive Error Field in Machining Space of Five-axis Machine Tool Based on S-shaped Specimen Family ”(machines-1622515)

Dear editors and reviewers:

Many thanks for the insightful comments and suggestions of the reviewers. Considering the reviewers' comments and suggestions, I have made in responses to the reviewers' questions and suggestions on an item by item basis. The amendments are made revisions to the manuscript using Track Changes. The following is the responses to the comments for “Prediction of Comprehensive Error Field in Machining Space of Five-axis Machine Tool Based on "S"-shaped Specimen Family”.

We appreciate for Editors/Reviewers’ warm work earnestly, and hope that the correction will meet with approval.

With best wishes

Yours sincerely

Round 2

Reviewer 1 Report

The article can be accepted for publication. The authors applied significant changes and corrections to the work. The authors' responses are correct